# Beyond Consensus: Mitigating the Agreeableness Bias in LLM Judge Evaluations

## Abstract

New Large Language Models (LLMs) become available every few weeks, and modern application developers confronted with the unenviable task of having to decide if they should switch to a new model. While human evaluation remains the gold standard, it is costly and unscalable. The state-of-the-art approach is to use LLMs as evaluators (*LLM-as-a-judge*), but this suffers from a critical flaw: LLMs exhibit a strong positive bias. We provide *empirical* evidence showing that while LLMs can identify valid outputs with high accuracy (i.e., True Positive Rate $> 96\%$), they are remarkably poor at identifying invalid ones (i.e., True Negative Rate $< 25\%$). This systematic bias, coupled with class imbalance, often leads to inflated reliability scores.

While ensemble-based methods like majority voting can help, we show that they are not good enough. We introduce an optimal minority-veto strategy that is resilient to missing data and mitigates this bias to a large extent. For scenarios requiring even higher precision, we propose a novel regression-based framework that directly models the validator bias using a small set of human-annotated ground truth data. On a challenging code feedback task over 366 high-school Python programs, our regression approach reduces the maximum absolute error to just $1.4\%$, achieving a $2\times$ improvement over the best-performing ensemble of 14 state-of-the-art LLMs.

## 1 Introduction

Recent advances in Generative AI have taken the world by storm, with leading companies releasing new large language models (LLMs) every few weeks. This means that application developers constantly have to decide if they should switch to a new model. While human evaluation remains the gold standard for accuracy, it is resource-intensive and limited by cost, time, and scalability.

This has caused developers to turn to automated evaluation. While benchmarking is straightforward for tasks with unambiguous correctness criteria, such as multiple-choice questions in MMLU (Hendrycks et al., 2021) or coding problems with unit-test oracles in HumanEval (Chen et al., 2021), benchmarking is significantly more challenging for a new class of open-ended problems where multiple distinct outputs can be correct, such as generating feedback for incorrect student programs from the IntroPython dataset (Sahai et al., 2023). In this paper, we investigate the challenges of using LLMs as judges for this emerging problem class, which is active research area.

The state-of-the-art automated benchmarking approach, "LLM-as-a-judge" (Zheng et al., 2023), suffers from high variance across different validator models, making it difficult to select a reliable judge (Wang et al., 2023). More critically, as observed in prior work (Thakur et al., 2024), we found even the latest LLM models exhibit a strong positive bias, i.e. they are proficient in identifying correct answers, but they perform poorly in identifying incorrect ones. Our analysis reveals a low True Negative Rate (TNR) across all models (§3), which leads to an overestimation of model precision. This low TNR undermines the reliability of LLM-based evaluation, especially since the proportion of invalid outputs is typically small. Furthermore, popular benchmarking frameworks such as Elo scoring (Zheng et al., 2023; Chiang et al., 2024), while cleverly using human preferences for relative model comparisons, do not reflect absolute model performance or evaluator reliability. As we can see in Figure 1, even high Elo-ranked models can exhibit poor TNR, suggesting that Elo rankings alone are insufficient to decide on the best model for a task.

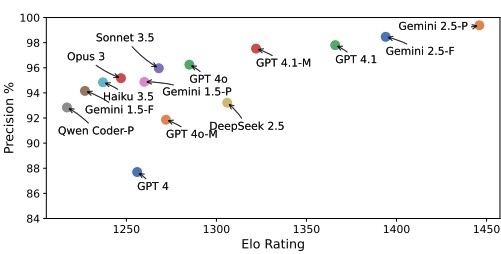 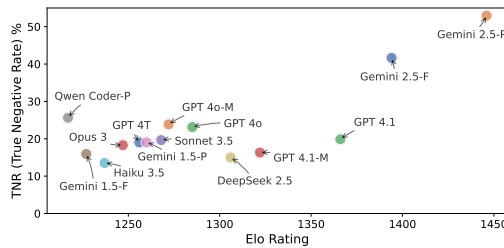

(a) Elo rating vs absolute precision of a generator.  (b) Elo rating vs True Negative Rate of a validator.

Figure 1: Correlation between LLM Elo ratings (Chiang et al., 2024) and their performance as generators and validators on high-school programming feedback.

A common strategy to mitigate individual validator bias is to use an ensemble of judges and rely on majority voting. However, we show that this approach is highly sensitive to data quality issues, such as missing values in validator outputs (§4). While it is possible to improve accuracy by manually repairing these data points, it is costly to do so. Furthermore, even with repaired data and novel voting strategies, the low TNR of individual validators imposes an upper bound on the accuracy of ensemble strategies.

Developers will often have access to the ground-truth data for a small number of (older) LLM models. Our key insight is that it is possible to exploit this data to mitigate the bias in TNR with a regression-based approach to avoid the expensive manual effort required to repair the validation data (§5). We demonstrate that our regression approach is robust to errors in the validation data and produces estimates that are more accurate than those obtained by an ensemble working with manually cleaned data.

In summary, our key contributions are as follows:
**1. Empirical quantification of evaluator bias**: Through a large-scale analysis of 14 LLMs as judges on a subjective task, we quantify previously observed bias, showing that while LLMs achieve high True Positive Rates (TPR) in classifying correct outputs (often $> 96\%$), their True Negative Rates (TNR) for identifying invalid outputs remain low (typically $< 25\%$).
**2. New dataset and code for subjective task**: To support reproducible research, we release an *extended dataset* building on Sahai et al. (2023), with 366 buggy programs, feedback generated by 14 LLMs, and validation judgments from all 14 models, along with human annotations for 6 of the generators. Associated code and analysis scripts will be made publicly available upon publication.
**3. Robust ensemble strategy**: We introduce a *minority-veto* ensemble strategy that is robust to data quality issues and outperforms standard majority voting.
**4. Novel regression-based methodology**: We propose a novel regression-based approach that, by leveraging a small set of human-annotated ground-truth data, jointly estimates a generator's precision and a validator's reliability (TPR/TNR), explicitly correcting for the validator's bias. With just five annotated datasets for calibration, our method reduces the maximum prediction error to $1.4\%$, a $2\times$ improvement over the best minority-veto ensemble.

To the best of our knowledge, we are the first to conduct a comprehensive study of a new class of open-ended problems with multiple correct answers. While existing "LLM-as-a-judge" approaches are known to suffer from significant positive bias, to the best of our knowledge, we are the first to quantify the bias as low TNR. We show that this bias can be mitigated with a minority-voting ensemble, and that the error can be further reduced with a regression-based method, with only a small amount of available ground truth data.

## 2 PROBLEM SETTING AND MOTIVATION

Building on these observations, we formalize the evaluation challenge our work aims to address. Specifically, we focus on tasks where reliable evaluation of LLM output is difficult due to the open-ended nature of the underlying problem.

Table 1: Human-annotated ground truth for six generators ($G_i$) and their overall precision ($g_i$).

| Generator | Valid | | | Invalid | | Precision |
|---|---|---|---|---|---|---|
| $G_i$ | TP | TP-E | TP-R | FP-I | FP-H | $g_i$ |
| **GPT 4** | 637 | 129 | 0 | 75 | 38 | 87.1% |
| **Opus 3** | 726 | 92 | 12 | 32 | 8 | 95.4% |
| **Gemini 1.5-P** | 988 | 103 | 51 | 82 | 6 | 92.8% |
| **GPT 4o** | 734 | 188 | 24 | 54 | 12 | 93.5% |
| **Qwen Coder-P** | 717 | 98 | 15 | 61 | 3 | 92.8% |
| **Deepseek 2.5** | 791 | 162 | 21 | 50 | 22 | 93.1% |

**Setup.** We consider a set of tasks $\mathcal{T}$, where each task $t$ has solutions that can be objectively classified as either valid or invalid. The core challenge is that the solution space is effectively unbounded, and no computationally efficient method exists to verify correctness.

As a motivating example, consider a typical introductory programming course (CS1). Given a problem description, test cases, and a student's buggy program, the task is to generate feedback that helps the student identify and fix their mistakes. In our work, $\mathcal{T}$ is a dataset of 366 incorrect Python program submissions for 69 different assignments (Sahai et al., 2023). Task $t$ is formally defined as: "Given a student's incorrect code, problem description, and test cases, generate *valid* feedback to help the student fix their mistake(s)."

For a task $t$, we use a generator $G_i$ from a set of LLMs $\mathcal{G}$ to generate an output $o$. A human expert labels each output as either *valid* ($H(t, o) = 1$) or *invalid* ($H(t, o) = 0$). By annotating all outputs from a generator $G_i$, we can compute its true precision $g_i$ as the fraction of valid outputs. However, manual annotation is resource-intensive and unscalable. Our goal is to estimate the precision $g_x$ for any new generator $G_x \in \mathcal{G}$ without requiring additional human annotation.

**Ground Truth Data.** Our evaluation is based on the high school Python programming assignment benchmark (Sahai et al., 2023), which includes human annotations of feedback generated by GPT 4. We extend this by annotating the outputs of five additional LLMs: Google's Gemini 1.5-Pro, Anthropic's Claude Opus 3, OpenAI's GPT 4o, Qwen Coder Plus, and Deepseek V2.5. For each of the 366 tasks, we prompted each LLM to generate feedback. All LLM prompts are included in the supplementary materials. Since feedback is granular and often tied to specific lines of code, a single submission can elicit multiple feedback items.

Human experts classified each feedback item, allowing us to compute the precision $g_i$ for each generator $G_i$, as shown in Table 1. We define precision as the proportion of valid feedback among all successfully generated items. To eliminate missing outputs, we re-ran the generation process upon failure. Annotating the additional models required an estimated 200 person-hours.

We extended the classification scheme from the original dataset to capture more nuanced feedback types. Valid feedback is categorized as: True Positive (TP) for identifying issues or suggesting fixes; TP-Extra (TP-E) for proposing code quality improvements; and TP-Redundant (TP-R) for providing non-essential information like code explanations. Invalid feedback is categorized as: False Positive-Incorrect (FP-I) for incorrect suggestions; and False Positive-Hallucination (FP-H) for feedback that refers to code elements that do not exist in the student's program.

**Challenges in Automated Validation.** Automating the validation of LLM-generated feedback is critical for scalability, but it is a deceptively complex task. It is not a simple pattern matching or syntactic check and is challenging for several reasons:

**1. Deep Semantic Reasoning:** A validator must interpret multiple artifacts—the student's buggy code, problem requirements, failing test cases, and natural language feedback—and reason about the student's *intent* versus the code's *actual* behavior to assess the feedback's validity.

**2. Compositional Complexity:** Validation is compositional, as the validator must assess both the problem diagnosis and the suggested solution. Feedback can be correct in its diagnosis but offer an invalid fix, or vice-versa, adding layers of complexity to the classification.

These challenges suggest that even sophisticated LLMs may struggle as validators, and lead to unreliable assessments. As we show, validators exhibit a systematic agreeableness bias: they reliably confirm correct feedback but frequently fail to reject incorrect feedback – a critical flaw for any

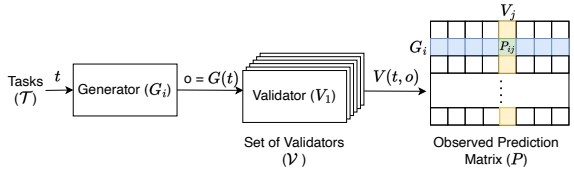

Figure 2: Workflow when using LLM to evaluate other LLM-generated feedback.

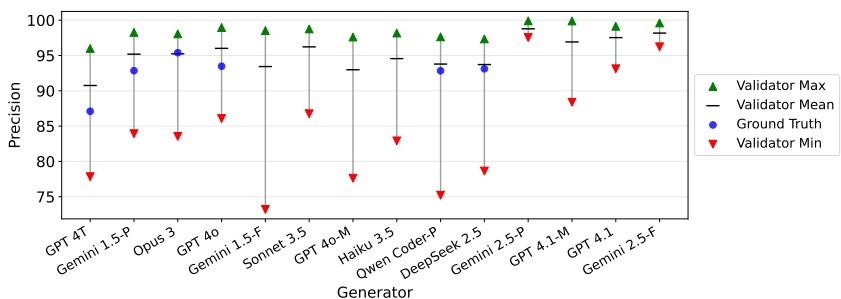

Figure 3: Predicted precision of generators by 14 validators.

automated benchmarking system that relies on them. We term this as "agreeableness" to describe the asymmetry in performance, characterized by a high True Positive Rate (TPR) but a low True Negative Rate (TNR).

While our study focuses on the single, challenging domain of code feedback generation, this task is representative of a broader class of open-ended problems where correctness is subjective and multiple solutions can be valid. Significant effort went into creating our ground-truth dataset (over 200 person-hours), which allowed for an in-depth analysis of evaluator bias that would be infeasible across multiple domains. While our findings provide a foundation for other subjective tasks, verification of its generalizability to other domains is left as future work.

## 3 LLM AS VALIDATORS

LLMs are increasingly used to evaluate the output of other LLMs (Zheng et al., 2023). This raises a natural question: *How reliable are LLMs at validating each other's outputs?*

To investigate this, we expanded our pool of models beyond the initial six with fully human-annotated outputs by including 8 additional LLMs: (i) GPT 4o-Mini, (ii) Gemini 1.5-Flash, (iii) Claude Haiku 3.5, (iv) Claude Sonnet 3.5, (v) GPT 4.1-Mini, (vi) GPT 4.1, (vii) Gemini 2.5-Pro, and (viii) Gemini 2.5-Flash. This brought the total to 14 models. Each LLM was tasked with generating feedback for 366 buggy programs, and all 14 models were subsequently used as validators to classify the outputs of all 14 generators. The validation workflow is depicted in Figure 2.

Figure 3 presents the precision estimates of each generator. Given that the models are sorted by release date, we can see that the precision and variance do not improve monotonically over time. Two key observations emerge:

**1. Systematic overestimation**: Validators consistently overestimate precision compared to human annotations. While the mean predictions align closely with the ground truth, they are consistently skewed higher, indicating a positive bias in validator judgments.

**2. High variance in estimates**: There is a wide spread in precision estimates, especially for less capable models. For instance, precision estimates for Gemini 1.5-Flash as a generator vary significantly, ranging from 73.2% to 98.6%. This high variance makes it difficult to determine which LLM is the best choice for a validator. The maximum error by any individual LLM-as-a-judge on our 6 annotated generators is 17.5% for the Gemini 1.5-Flash generator.

To better understand validator reliability, we computed the True Positive Rate ($v_j^+$) and True Negative Rate ($v_j^-$) for the six generators with human-labeled outputs. As shown in Figure 4, we found

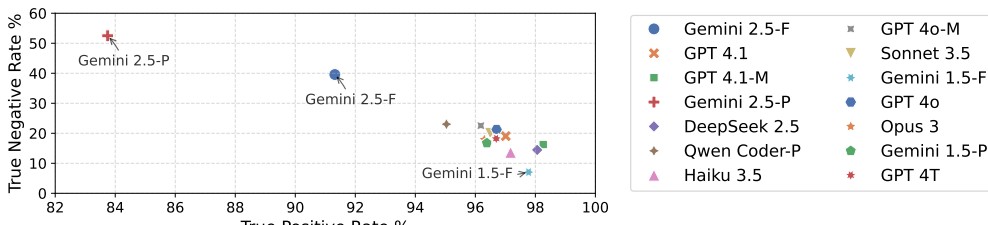

Figure 4: Agreeableness bias in LLM validators: high TPR ($v_j^+ \geq 96\%$) but low TNR ($v_j^- \leq 25\%$).

a striking disparity: while True Positive Rates consistently exceeded 96% across most validators, True Negative Rates were much lower — typically below 25%. This indicates that LLMs are far better at recognizing correct outputs than detecting mistakes, a tendency to be over-agreeable. This discrepancy is often masked by high overall accuracy, which is skewed due to the small fraction of invalid outputs in the entire dataset (about 7.5%, see Table 1).

Even flagship models like Gemini 2.5-Pro, while excellent as generators, struggle as validators. Its highest TNR (52.3%) comes at the cost of the lowest TPR (83.4%) among all models, highlighting that a model's generation strength does not guarantee its validation reliability.

This unreliability is further compounded by the fact that many LLM validators fail to provide output in the expected format. Around 9.4% of the validator outputs were left unlabeled due to missing or mismatched fields in the JSON output (refer to Appendix B.2 for more details). For example, if the feedback generated by the validator does not align with the corresponding feedback from the generator, it results in a "Missing Feedback" error type. This significant gap in validation coverage raises further questions about the dependability of individual LLMs as judges. This indicates that while LLMs can provide some insights into the performance of other LLMs, their high variance and low True Negative Rate (TNR) makes them unreliable to be used as a sole judge for benchmarking. This motivates exploring aggregator methods to mitigate individual model biases, which we discuss in the next section.

## 4 Ensemble of LLM Validators

To mitigate the unreliability of individual LLMs, a natural approach is to employ an ensemble (Lu et al., 2024; Jiang et al., 2023). The standard strategy is to use a *Simple Majority* consensus, which assigns a label as "valid" if the majority of validators agree and "invalid" otherwise. With a threshold of 8 (out of 14) validators, this strategy reduces the maximum error on our six annotated generators from 17.5% for the worst individual LLM to 12.2%. However, this approach is highly sensitive to missing data. After applying data repair techniques to reduce missing values from 9.4% to 3.2%, the maximum error for *Simple Majority* drops to 5.7%, highlighting its dependency on data completeness. These repairs involved three main steps: first, label standardization to unify validator outputs (e.g., mapping "incorrect" to 'invalid'); second, fuzzy string matching (Levenshtein distance) to align feedback items; and third, for feedback associated with line number ranges (e.g., '2-3'), we match the starting line number if an exact range match is not found. The scripts for these repairs will be released in our code repository.

Motivated by the low TNR of individual validators, we propose a *Minority Veto* strategy, which marks an output as "invalid" if at least $n$ validators agree, thus empowering a small minority to override the otherwise agreeable majority. As shown in Figure 5, a minority veto with just $n = 4$ votes decisively outperforms other methods, achieving the lowest maximum error of 2.8% after data repair. It is important to note that the choice of this optimal threshold ($n = 4$) itself requires a calibration set with ground-truth labels, and therefore, just like our regression approach, the optimized ensemble also leverages the available human annotations. This strategy strikes a superior balance between a high True Positive Rate (95.5%) and an improved True Negative Rate (31.6%) compared to both individual validators and the *Majority Consensus* strategy, which has a TNR of only 17.4%.

Crucially, our *Minority Veto* strategy is robust to the missing data that plagues the standard *Majority Consensus*. As shown in Figure 5, its performance is largely unaffected by the presence or repair

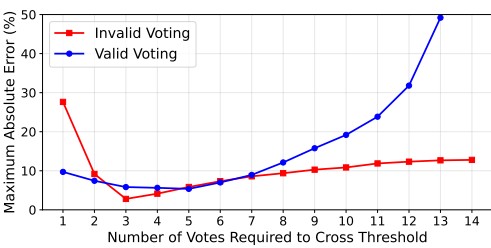 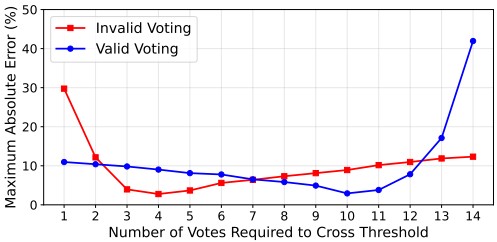

(a) Original dataset, with $9.4\%$ missing values

(b) After fixing issues, $3.2\%$ missing values remain

Figure 5: Valid voting strategies, which assign "valid" label based on threshold, mirror invalid voting after fixing missing values.

of missing values, making it a more reliable and practical approach. In contrast, a *Super Majority* strategy (e.g., requiring 10 of 14 votes for "valid") performs slightly worse. While this is conceptually equivalent to a minority veto, it defaults to "invalid" on missing data, which penalizes it heavily in a dataset with few invalid labels.

While ensemble methods, particularly the *Minority Veto* strategy, demonstrate the value of collective judgment, their ability to improve the True Negative Rate is still fundamentally limited by the biases of the individual models. Achieving even higher accuracy requires explicitly modeling and calibrating for the systemic validator bias. This motivates our regression-based approach.

## 5 REGRESSION-BASED BIAS CORRECTION

While ensemble methods offer a stronger baseline over single validators, they are fundamentally limited by the inherent biases of individual LLM judges. To achieve higher accuracy, we propose a regression-based framework that directly models and corrects for validator bias.

Our approach is motivated by a practical observation: while annotating outputs for every new generator is unscalable, evaluation pipelines often have access to a small set of human-annotated ground truth data for a few models. Our key idea is to leverage this "calibration set" to estimate the reliability (TPR and TNR) of each validator. These estimates can then be used to predict the precision of any new, un-annotated generator.

This method assumes that a validator's TPR and TNR are stable characteristics that do not vary significantly across different generators. While this is a strong assumption, our empirical results show that this framework is effective, achieving precision estimates that significantly outperform even the best ensemble strategies.

The key insight is that instead of relying on an aggregate of votes, we can model the interactions between generators and validators. We assume each generator $G_i$ has a precision $g_i$, and each validator $V_j$ has a characteristic True Positive Rate ($v_j^+$) and True Negative Rate ($v_j^-$). By leveraging a small set of human-annotated outputs, we can solve for these parameters. This allows us to correct for the systemic over-agreeableness of LLM judges and derive more accurate precision estimates for all generators, including those for which we have no annotated data.

### 5.1 METHODOLOGY

For a generator $G_i \in \mathcal{G}$ and a validator $V_j \in \mathcal{V}$, the probability that validator $V_j$ will label an output from generator $G_i$ as "valid" is given by:

$$\widehat{\mathrm{P}}_{ij} = \widehat{g_i} \cdot \widehat{v_j^+} + (1 - \widehat{g_i}) \cdot (1 - \widehat{v_j^-}) \tag{1}$$

This equation represents the sum of two probabilities: the generator was correct and the validator agreed (a true positive validation), or the generator was incorrect and the validator also made a mistake (a false positive validation). Our goal is to estimate the generator precisions ($\widehat{g_i}$), validator TPRs ($\widehat{v_j^+}$), and validator TNRs ($\widehat{v_j^-}$) that best explain the observed validation matrix P, where each

cell $P_{ij}$ represents the empirical fraction of "valid" labels assigned by validator $V_j$ to generator $G_i$'s outputs.

This becomes a regression problem where we estimate $|\mathcal{G}| + 2|\mathcal{V}|$ variables using the $|\mathcal{G}| \times |\mathcal{V}|$ data points in the matrix P. In our case, we estimate 42 variables (14 generator precisions and 28 validator parameters) from 196 observations (14 generator precision predictions by 14 validators). We formulate an optimization problem to minimize a loss function that combines two components: a prediction loss and a calibration loss.

The prediction loss, $\mathcal{L}_{pred}$, measures the difference between the observed and estimated validation outcomes using binary cross-entropy:

$$\mathcal{L}_{pred} = \frac{-1}{|\mathcal{G}||\mathcal{V}|} \sum_{i,j} \left[ P_{ij} \log \widehat{P}_{ij} + (1 - P_{ij}) \log (1 - \widehat{P}_{ij}) \right] \qquad (2)$$

However, minimizing $\mathcal{L}_{pred}$ alone is insufficient, as it often converges to solutions that reflect the inherent biases of the validators, resulting in systematic overestimation of precision. To mitigate this, we introduce a calibration loss, $\mathcal{L}_{cal}$, which anchors our model by penalizing deviations from known ground-truth values for the subset of generators, $\mathcal{H} \subset \mathcal{G}$, for which we have human annotations:

$$\mathcal{L}_{cal} = \lambda_g \sqrt{\frac{1}{|\mathcal{H}|} \sum_{G_i \in \mathcal{H}} (g_i - \widehat{g}_i)^2} + \lambda_{v^+} \sqrt{\frac{1}{|\mathcal{V}|} \sum_{V_j \in \mathcal{V}} (v_j^+ - \widehat{v}_j^+)^2} + \lambda_{v^-} \sqrt{\frac{1}{|\mathcal{V}|} \sum_{V_j \in \mathcal{V}} (v_j^- - \widehat{v}_j^-)^2} \quad (3)$$

The total loss is $\mathcal{L} = \mathcal{L}_{pred} + \mathcal{L}_{cal}$. The weights $\lambda_g, \lambda_{v^+}, \lambda_{v^-}$ control the influence of the ground-truth data. Through grid search, we found that placing higher weight on generator precision ($\lambda_g = 2$) and True Negative Rate ($\lambda_{v^-} = 10$) relative to True Positive Rate ($\lambda_{v^+} = 1$) yielded the best results. This confirms our hypothesis that correcting for the low TNR is critical.

We solve this optimization problem using the L-BFGS-B algorithm (Zhu et al., 1997), a quasi-Newton method that efficiently handles a large number of free variables by approximating the Hessian matrix using limited memory, with box constraints to ensure all estimated parameters remain within $[0, 1]$.

To estimate the precision of a new, unannotated generator, we first evaluate its outputs using our existing suite of validators, thereby appending a new row to the prediction matrix P. We then re-run the optimization to solve for the precision of the new generator alongside the existing parameters. We use the available ground-truth data to anchor the validator bias terms ($\widehat{v}_j^+, \widehat{v}_j^-$), effectively transforming the evaluation of a new model into a straightforward estimation problem.

## 5.2 EVALUATION

To evaluate our regression approach, we simulate how its performance improves as more ground-truth data is made available for calibration. We have 6 generator datasets with full human annotations. We conduct an experiment where we train our regression model using a varying number of these annotated datasets, which we will refer to as the calibration set, and test its accuracy on the remaining held-out sets.

Specifically, for each size $s \in \{0, \ldots, 5\}$, we create calibration sets by enumerating all combinations of $s$ annotated datasets. The model is trained on the ground-truth calibration set, and evaluated on the remaining $6 - s$ datasets. For each $s$, we report the mean Maximum Absolute Error across all combinations. Since numerical optimization may not converge to the global optimum, we perform 25 regression runs with different initializations and select the one with the lowest training loss.

For each run, generator precisions ($\widehat{g}_i$) were initialized to their row means in P while validator TPRs/TNRs were set to 1 with small random delta to explore the solution space. All experiments were conducted on a machine with an AMD EPYC 7B12 (32-core, 32GB RAM, Debian 12). The complete pipeline, encompassing all calibration set combinations and 25 runs per configuration, took approximately 1.5 hours to complete.

Figure 6a shows the distribution of the error in estimating generator precision when evaluated against various held out test set of $(6 - s)$ size. Figure 6b compares the maximum absolute error of our

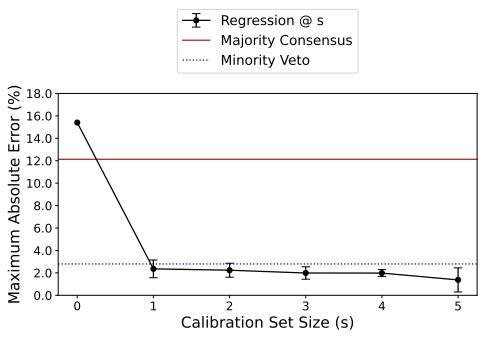

(a) Spread of MaxAE across 25 runs and calibration sets, on original dataset with missing values

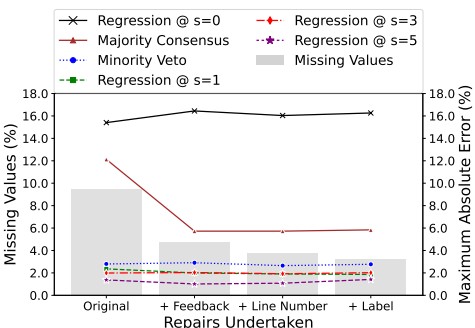

(b) Mean of MaxAE of different methods, across data repair strategies

Figure 6: Our regression approach, which leverages ground-truth data, outperforms other methods as more annotated datasets are included ($s \geq 1$) and is resilient to missing values.

regression method against the best individual LLM judge and the best ensemble strategy. Without any ground truth ($s = 0$), the regression performs poorly, with a maximum error of $15.8\%$, which is worse than other methods. This highlights that the model requires calibration.

The introduction of just a single annotated dataset ($s = 1$) dramatically improves performance, matching the best ensemble method. As more annotated data is incorporated ($s \geq 2$), the regression model's error drops further, outperforming all other approaches. With five annotated datasets for calibration ($s = 5$), the maximum error falls to just $1.4\%$ on the held-out test set, a $2\times$ improvement over the best ensemble. Moreover, as seen in Figure 6b, missing data has a negligible impact on regression performance, as it can leverage the available ground-truth data to calibrate the model.

Based on these results, our regression framework can provide better precision estimates for LLMs, by explicitly modeling and calibrating for validator biases. By using small amounts of human annotation, this approach leverages the strengths of both human judgment and LLM capabilities, resulting in a more reliable benchmarking process.

## 6 RELATED WORK

**LLM-as-Judge.** The usage of LLMs for evaluating other LLMs is now widespread (Gu et al., 2024). Prior work has identified various biases, such as preferences for position, verbosity, and a general leniency or "self-enhancement" bias where LLMs rate their own outputs favorably (Zheng et al., 2023). Our work differs by framing this leniency as a quantifiable "agreeableness bias", characterized by an imbalance between high TPR and low TNR. While relatively ranking LLMs in terms of strength is popular (Chiang et al., 2024), we make use of a binary (True/False) setup instead as relative ranking can obscure the absolute correctness or precision of generator outputs, as seen in Figure 1a.

**Ensemble Methods.** Ensemble approaches are a natural solution to mitigate individual model biases. While simple majority voting is common, recent work has explored more sophisticated strategies like generative fusion (Jiang et al., 2023) and selecting diverse sub-ensembles (Tekin et al., 2024). However, these methods remain fundamentally constrained by the systematic biases of their constituent models. As demonstrated in §4, standard voting is insufficient for addressing the low TNR that characterizes LLM judges, and even our optimized minority-veto strategy is limited by this inherent flaw.

**Calibration and Latent Trait Models.** Our regression framework is conceptually related to classic models for aggregating noisy labels, such as the EM Algorithm (Dawid & Skene, 1979) and Item Response Theory (IRT) (Lord, 2012). While IRT is powerful for relative ranking, its latent variables do not directly map to an absolute, interpretable metric like precision. Our model is tailored to this estimation task by explicitly modeling generator precision ($g_i$) as a variable, similar to calibration techniques, which aim to align model scores with correctness probabilities (Guo et al., 2017). How-

ever, existing methods typically address general overconfidence rather than the specific, systematic low-TNR problem that we identify. Our regression-based approach differs by using a ground-truth calibration set to explicitly model and correct for this bias, shifting from general confidence scaling to targeted bias correction. We refer the reader to our Appendix E for a more detailed discussion.

# 7 LIMITATIONS AND FUTURE WORK

Our voting ensemble and regression framework offers a robust method for estimating LLM precision, but our assumptions leave scope for future work.

**Dependence on Annotations:** Our regression framework's accuracy is anchored by a ground-truth calibration set. While accuracy improves dramatically with even a single annotated generator dataset ($s \geq 1$), our current method assumes annotation of all the outputs by this generator. A more efficient approach would be to focus human effort on the most informative outputs, such as those where validators exhibit high disagreement. Future work could explore active learning strategies to systematically identify these high-impact data points for labeling, potentially achieving comparable accuracy with significantly less annotation effort.

**Generalizability to other domains:** Our work focuses on code feedback, a domain with a relatively objective ground truth. The underlying problem of low TNR in LLM judges likely exists in other domains, but the applicability of our solution for other subjective tasks, where human judges could also potentially disagree among each other, is left to future work. Our regression model could be adapted to handle the increased ambiguity of subjective tasks, for instance by incorporating models of inter-rater disagreement among human annotators into the loss function.

**Model Simplifications:** Our model estimates aggregate performance by assuming that a generator's precision ($g_i$) and a validator's TPR/TNR ($v_j^+$, $v_j^-$) are static properties. While this simplification is effective for overall benchmarking, these values could in practice vary with problem difficulty. For example, a validator's ability to detect an invalid output might depend on the type of error in code or the generator that produced it. Future work could build upon our framework by developing hierarchical models that capture problem/model-specific characteristics.

**Evaluation Metric.** This study estimates the generator's precision ($g_i$), defined as the proportion of valid outputs among all generated outputs. While precision provides a straightforward measure of performance, it does not fully account for the overall quality or coverage of the generator's feedback. Future research could explore metrics that evaluate aspects such as relevance, usefulness, or recall of the generated outputs.

**Scalability of the Prediction Matrix:** Our regression methodology requires a full, dense prediction matrix – every generator must be evaluated by every validator – resulting in quadratic complexity ($|\mathcal{G}| \times |\mathcal{V}|$). This can be computationally intensive for large-scale comparisons involving many models. While this is a one-time cost and does not repeat for each new model that we benchmark, it may pose a limitation. The investigation of methods to train the model on a sparse prediction matrix, where only a subset of generator-validator pairs are run, is left as future work.

# 8 CONCLUSION

In this paper, we quantify the "agreeableness bias" of LLM-as-a-judge on a challenging task of programming feedback on 366 incorrect high school programs. Our large-scale study empirically shows that even state-of-the-art validators, while great at recognizing valid outputs (TPR $> 96\%$), are poor at identifying invalid ones (TNR $< 25\%$), which leads to overestimation of generator precision by the validators. We show that while a *Minority Veto* ensemble can mitigate this bias better than standard majority voting, its accuracy remains fundamentally limited by the poor TNR of individual LLMs. To overcome this limitation, we propose a regression-based framework that explicitly models and corrects for validator bias. With only a small set of human-annotated data, our method can accurately estimate generator precision without requiring further manual labeling. Our approach can reduce the maximum estimation error to just 1.4% - a twofold improvement over the best-performing ensemble, demonstrating a more reliable and scalable LLM evaluation framework. We leave the investigation of the effectiveness of our approach for other domains, including those with human-generated content, as future work.

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

## REPRODUCIBILITY

To ensure reproducibility, we will release our code, data, and plotting scripts upon publication. This includes our dataset of feedback from 14 LLMs on 366 buggy programs, with human annotations for 6 of the generator outputs. Our methodologies for ensemble strategies and the regression framework are detailed in Section 4 and Section 5. The experimental setup, including optimization parameters, is described in Section 5. This appendix provides further details, including the prompts used for all LLMs and an analysis of missing data.

## LLM USAGE STATEMENT

We utilized Large Language Models (LLMs) to improve the conciseness and check for grammatical correctness of the text. The core research ideas, experimental design, analysis, conclusions and initial writeup are entirely our own. All of the generated suggestions were reviewed by us before incorporating it into the draft.

## A   LLM PROMPTS

### A.1   GENERATOR PROMPT

We use the following prompt to generate code and feedback for the high-school programming tasks:

---

**System Role**

*Given an incorrect student Python program, fix the program with minimal changes and generate line-numbered feedback. The following inputs are provided:*

- **Problem description**
  [QUESTION]

- **Prefix**
  [PREFIX]

- **Student's incorrect code**
  [STUDENT CODE]

- **Suffix**
  [SUFFIX]

- **Test Cases**
  [TEST CASES]

*Make changes **only** to the student's incorrect code such that it passes **all** the test cases. STRICTLY output a JSON in the following format:*

```
{
  "correct_code": "The student code with minimal changes to pass
     all test cases",
  "feedbacks": [
    {
      "line_number": "line number where the mistake occurs",
      "feedback": "the feedback you would like to give the student
         to fix their mistake"
    }
  ]
}
```

---

## A.2 VALIDATOR PROMPTS

We make use of Chain-of-Thought with Few Shot examples for validating LLM generated feedback by another LLM judge.

- **Chain-of-Thought (CoT):** The validator is prompted to follow a step-by-step reasoning process: first, identify mistakes in the student's code; second, compare it with the provided fix; and third, evaluate the feedback. The `"analysis"` field forces the model to state its reasoning

- **Few-Shot Example:** The prompt includes a complete example demonstrating the task. This one-shot example, with a user query and an ideal assistant response, helps the validator understand the expected output format.

---

**System Role**

**Task:** *Your goal is to assess the validity of feedback provided by a Teaching Assistant (TA) on a student's incorrect Python program.*

*First, analyze the student's code to identify specific mistakes.*

*Then, compare it with the TA's fixed version and pinpoint the changes made to correct those mistakes.*

*Finally, examine each feedback line from the TA and determine whether it accurately addresses the mistake.*

**For each feedback line:**
- Classify it as either `"valid"` or `"invalid"`
- Justify your classification with a brief analysis

**Output Format:** Provide your analysis in the following JSON format:

```
{
  "mistakes": [],
  "fixes": [],
  "feedback_lines": [
    {
      "line_number": <integer>,
      "feedback": <string>,
      "analysis": <string>,
      "classification": "valid" | "invalid"
    }
  ]
}
```

---

**User Role**

**Problem Description:**

You have two variables, x and y, each containing a random integer value. Your task is to write a piece of code that will exchange the values of these two variables. This means that the value initially held by x should be in y and vice versa.

**Buggy Student Program:**

```
x = "newy"
y = "newx"
x, y = x, y
print(x)
print(y)
```

**Fixed Code Generated by the Teaching Assistant:**

```
x, y = y, x
print(x)
print(y)
```

**Feedback Lines by the Teaching Assistant (may be invalid):**

```
[
  {
    "line_number": 1,
    "feedback": "You don't need to assign new strings to x and y
        before swapping their values. You can directly swap the
        values using the syntax \"x, y = y, x\"."
  }, ...
]
```

**Test Case Results:**

```
[
  {"expression": "x == newx", "success": false},
  {"expression": "y == newy", "success": false}
]
```

**Assistant Role**

**Assistant's Response:**

```
{
  "mistakes": ["The student has made a mistake when trying to swap
      the values of x and y"],
  "fixes": ["Directly swap the values instead, using the syntax \"x
      , y = y, x\"."],
  "feedback_lines": [
    {
      "line_number": 1,
      "feedback": "You don't need to assign new strings to x and y
          before swapping their values. You can directly swap the
          values using the syntax \"x, y = y, x\".",
      "analysis": "The TA has correctly identified the mistake made
          by the student while swapping the values and its fix.",
      "classification": "valid"
    }, ...
  ]
}
```

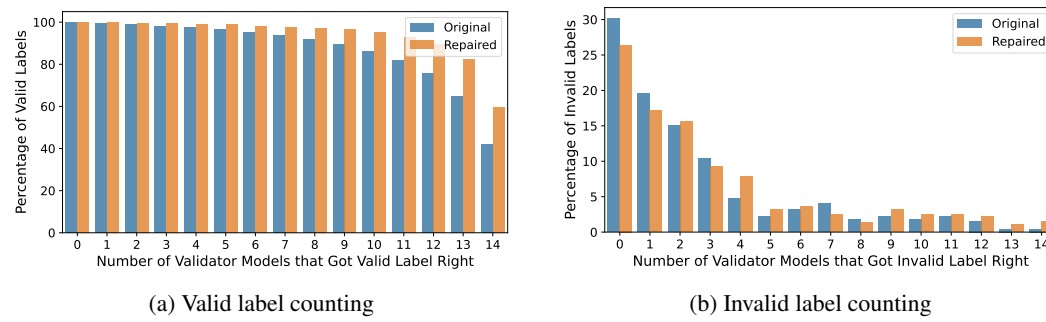

(a) Valid label counting

(b) Invalid label counting

Figure 7: Number of validator models that got the label right. None of the LLM validators could identify around 28% of the invalid labels even after repairing some of the missing values.

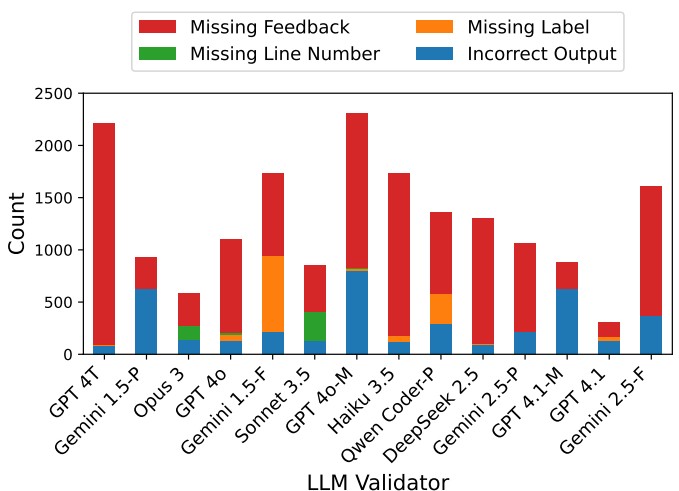

Figure 8: Number of missing value failures for each LLM validator when classifying generator outputs.

# B LLM VALIDATOR

## B.1 IDENTIFYING VALID AND INVALID LABELS

Figure 7 shows the cumulative distribution of valid and invalid labels across 14 LLM validators. Approximately 60% of the valid labels were correctly identified by all 14 validators. Conversely, the agreement among validators for invalid labels was significantly lower, with only 2% correctly identified by all. Notably, 28% of the invalid labels were missed by every validator, even after the repair process. This highlights the challenge of identifying invalid labels, as even the best-performing validators struggled with the task.

## B.2 MISSING VALUES

Figure 8 shows the distribution of missing values across various LLM validators. About 18K of 193K (9.4%) LLM generator outputs were not labeled due to LLM validation failures. Relatively larger models, such as Opus 3 and GPT 4.1 , had fewer missing values, while smaller models like GPT 4T and GPT 4o-M exhibited a higher rate of missing values, indicating that larger models may be more reliable in generating consistent output.

### B.3 REPAIR MISSING VALUES

We repair some of the missing values in the dataset by applying fuzzy matching on the feedback, line-number and labels. As explained in our main paper, a large number of LLM validator outputs could not be aligned with the LLM generated feedback due to issues in reproducing the original feedback, line number, or binary label (valid/invalid). Instead of an exact match, we make use of the FuzzyWuzzy library (Cohen, 2011) to match the feedback referenced by the LLM validator with LLM generator's feedback. If the match score is above 85% threshold, we declare the match a success. Similarly, we fix issues with line numbers, where the validator could have used an incorrect key (e.g, "line_num" instead of 'line_number') or a different format (e.g., "line 3" instead of "3"). We also fix the binary labels, where the validator could have invented a new label (e.g., ''partially valid" instead of the expected "valid" and "invalid"). After applying these repairs, the missing values reduce drastically from 9.4% to 3.2%. Following which, *Simple Majority* consensus ($m = 8$ valid voting ensemble) had a major improvement in its max error rate from 12.2% to 5.7%.

## C REGRESSION DERIVATION

In this section, we provide the derivation for the main regression equation used to calibrate the precision of generators and the reliability of validators.

### C.1 PROBABILISTIC MODEL

Given a generator $G_i \in \mathcal{G}$ to generate an output $o$ for a task $t$ and a validator $V_j \in \mathcal{V}$, we assume the following probabilistic behavior:

$$\Pr(V_j(o) = 1 \mid H(t, o) = 1) = v_j^+, \tag{4}$$

$$\Pr(V_j(o) = 0 \mid H(t, o) = 0) = v_j^-, \tag{5}$$

where $o$ is an output generated by generator $G_i$, and $H(t, o)$ is the human-annotated ground truth label, such that $H(t, o) = 1$ if the output is *valid* and $H(t, o) = 0$, otherwise.

The probability of $G_i$ producing a valid output is defined as its precision, denoted by $g_i$:

$$\Pr(H(t, o) = 1) = g_i. \tag{6}$$

### C.2 VALIDATOR-GENERATOR INTERACTION

The probability that a validator $V_j$ will label an output from generator $G_i$ as "valid" is the sum of two cases:

1. Generator output is valid, and the validator correctly identifies it as valid.
2. Generator output is invalid, but the validator incorrectly labels it as valid.

Formally, this probability is expressed as:

$$\Pr(V_j(o) = 1) = \Pr(H(t, o) = 1)\Pr(V_j(o) = 1 \mid H(t, o) = 1) \\ + \Pr(H(t, o) = 0)\Pr(V_j(o) = 1 \mid H(t, o) = 0) \tag{7}$$

Substituting our notation:

$$\Pr(V_j(o) = 1) = g_i \cdot v_j^+ + (1 - g_i)(1 - v_j^-). \tag{8}$$

### C.3 EMPIRICAL ESTIMATION

Define $\mathrm{P}_{ij}$ as the empirical fraction of outputs from generator $G_i$ labeled valid by validator $V_j$. Given a set of outputs $\mathcal{O}_i$ from generator $G_i$, this fraction is computed as:

$$\mathrm{P}_{ij} = \frac{1}{|\mathcal{O}_i|} \sum_{o \in \mathcal{O}_i} \mathbb{1}(V_j(o) = 1). \tag{9}$$

We approximate $\mathrm{P}_{ij}$ by its expectation, yielding the following equation:

$$
\begin{aligned}
\widehat{\mathrm{P}}_{ij} &= \mathbb{E}[\mathrm{P}_{ij}] \\
&= \mathbb{E}\left[\frac{1}{|\mathcal{O}_i|}\sum_{o\in\mathcal{O}_i}\mathbb{1}(V_j(o)=1)\right] \\
&= \frac{1}{|\mathcal{O}_i|}\sum_{o\in\mathcal{O}_i}\mathbb{E}\left[\mathbb{1}(V_j(o)=1)\right] \\
&= \frac{1}{|\mathcal{O}_i|}\sum_{o\in\mathcal{O}_i}\Pr(V_j(o)=1) = \Pr(V_j(o)=1) \\
&= g_i\cdot v_j^+ + (1-g_i)(1-v_j^-).
\end{aligned}
\tag{10}
$$

### C.4   LOSS FUNCTION

To estimate the parameters $g_i$, $v_j^+$, and $v_j^-$, we minimize a loss function consisting of two components:

**Prediction Loss** ($\mathcal{L}_{pred}$): This measures the discrepancy between observed validator outputs and our model predictions, using binary cross-entropy:

$$
\mathcal{L}_{pred} = \frac{-1}{|\mathcal{G}||\mathcal{V}|}\sum_{i,j}\left[\mathrm{P}_{ij}\log\widehat{\mathrm{P}}_{ij} + (1-\mathrm{P}_{ij})\log\left(1-\widehat{\mathrm{P}}_{ij}\right)\right]
\tag{11}
$$

**Calibration Loss** ($\mathcal{L}_{cal}$): This anchors the model to known ground-truth annotations for a subset of generators, $\mathcal{H}\subseteq\mathcal{G}$:

$$
\begin{aligned}
\mathcal{L}_{cal} \quad = \quad & \left(\lambda_g\sqrt{\frac{1}{|\mathcal{H}|}\sum_{G_i\in\mathcal{H}}(g_i-\widehat{g_i})^2}\right) \\
& + \left(\lambda_{v^+}\sqrt{\frac{1}{|\mathcal{V}|}\sum_{V_j\in\mathcal{V}}(v_j^+-\widehat{v}_j^+)^2}\right) \\
& + \left(\lambda_{v^-}\sqrt{\frac{1}{|\mathcal{V}|}\sum_{V_j\in\mathcal{V}}(v_j^--\widehat{v}_j^-)^2}\right)
\end{aligned}
\tag{12}
$$

The overall optimization problem becomes:

$$
\min_{\{g_i,v_j^+,v_j^-\}}\left(\mathcal{L}_{pred}+\mathcal{L}_{cal}\right)
\tag{13}
$$

This formulation explicitly corrects for the inherent biases of validators, particularly addressing their low True Negative Rates.

## D   REGRESSION ADDITIONAL RESULTS

Our goal is to estimate the precision $g_x$ of a new LLM $G_x$ with minimal human annotation. As shown in our main paper, ensemble-based methods are insufficient for this task. We propose a regression-based approach that leverages two key insights: (i) LLM validators are systematically biased, often over-estimating the correctness of invalid outputs (i.e., exhibiting low TNR), and (ii) a small set of ground-truth annotations can be used to calibrate the model and correct for these biases.

We solve the optimization problem using the BFGS algorithm (Fletcher, 2000) with a convergence tolerance of $10^{-6}$ on the prediction loss. The parameters are initialized as follows: generator precisions $\widehat{g}_i$ are set to the mean of their corresponding row in the prediction matrix P, while validator

TPRs $\widehat{v}_j^+$ and TNRs $\widehat{v}_j^-$ are initialized to 1.0. To mitigate the risk of local minima, we perform 25 runs with small random offsets added to the initial values and select the run with the lowest training error.

### D.1 Regression Error

To evaluate our regression approach, we simulate how its performance improves as more ground-truth data is made available for calibration. We have 6 generator datasets with full human annotations. We conduct an experiment where we train our regression model using a varying number of these annotated datasets, which we refer to as the calibration set, and test its accuracy on the remaining held-out sets.

Specifically, for each size $s \in \{0, 1, 2, 3, 4, 5\}$, we create calibration sets by enumerating all combinations of $s$ annotated datasets. The model is trained on the ground-truth data from the calibration set and evaluated on the remaining $6 - s$ datasets. For each $s$, we report the Maximum Absolute Error for each combination.

Figure 6a compares the maximum absolute error of our regression method against several baselines, evaluated against the held out test set of size $(6 - k)$, for the original dataset with 9.4% missing values. Relying on a single LLM judge is unreliable; as shown in the main paper, their performance varies significantly, and a seemingly good result can be due to chance, given their low True Negative Rates. Ensemble methods offer more consistency. A simple mean of all validator predictions provides a surprisingly strong baseline. However, a more sophisticated strategy like *Minority Veto* improves performance further by enhancing the effective TNR, achieving a maximum error of 2.3% on the held-out test set.

Our regression approach, when calibrated with ground-truth data, significantly outperforms these baselines. Without any ground truth ($s = 0$), the uncalibrated model performs poorly, with a maximum error of 15.8%. However, with just a single annotated dataset for calibration ($s = 1$), the error drops dramatically, matching the best ensemble method. As more annotated data is incorporated ($s \geq 4$), the regression model's error falls further. With five annotated datasets ($s = 5$), the maximum error is just 1.4%, a $2\times$ improvement over the best ensemble. Notably, with four or more annotated datasets, even the worst-performing calibration set combination for our model yields a lower maximum error than the best ensemble strategy. In the optimal case, our regression achieves a maximum error of only 0.2% on the held-out test set.

These results demonstrate that even with a small set of ground truth annotations, we can effectively mitigate validator bias and significantly improve precision estimates.

## E Related Work - Expanded

**Human Evaluation**. Human evaluation has traditionally been the gold standard for assessing language model performance, especially for inherently subjective tasks such as translation, summarization, and open-ended question answering (Brown et al., 2020). In these evaluations, human judges rate the quality of model outputs based on specific criteria, providing reliable insights into the model's capabilities. Another approach involves collecting human preferences between two or more model outputs in head-to-head comparisons (Zheng et al., 2023). However, these approaches lack scalability, particularly with the rapid release of newer models requiring frequent evaluation.

**LLM-as-a-Judge**. The use of Large Language Models as evaluators has gained significant traction for assessing open-ended text generation quality, often achieving over 80% agreement with human judgments (Zheng et al., 2023). However, recent work has revealed critical vulnerabilities within these systems. Studies indicate that even state-of-the-art LLMs suffer from poor recall, failing to identify the majority of errors made by other LLMs on subjective tasks (Kamoi et al., 2024). LLM judges has earlier been reported to exhibit systematic biases including positional bias where response order and persuasive repetitive language influences judgments (Wang et al., 2023; Pezeshkpour & Hruschka, 2024); and more critically, what we term "agreeableness bias," or a tendency to accept invalid outputs as correct. Simple adversarial inputs that suggest deep reasoning such as "Thought process:" or even non-word symbols can trigger false positive rewards exceeding 80% for state-of-

the-art models including GPT 4o and Claude Sonnet 4 (Zhao et al., 2025). We are the first to show with data that the TNR for existing LLMs are often below 25%.

An alternative to using off-the-shelf LLMs as judges is to fine-tune a verifier model, which has been shown to improve reliability on datasets like GSM8K (Cobbe et al., 2021). However, this approach is costly, prone to overfitting, and yields limited benefits with scarce training data. Furthermore, a verifier trained on one dataset often lacks generalizability, requiring retraining for new evaluation tasks.

**Ensemble Methods**. Ensemble approaches represent a natural solution to mitigate individual LLM biases by leveraging diverse model strengths. While simple jury voting schemes are common, recent work has explored more sophisticated strategies, as reviewed in Chen et al. (2025). LLM-BLENDER (Jiang et al., 2023) combines pairwise ranking with generative fusion, using specialized comparison methods to distinguish candidate outputs and generate a unified high quality response. The Mixture-of-Reasoning-Experts (MORE) framework (Si et al., 2023) creates specialized experts for different types of reasoning and leverages inter-expert agreement for both answer selection and abstention, achieving superior generalizability and question answering performance.

Mirror-Consistency Huang et al. (2024a) enhances Self-Consistency by incorporating a "reflective mirror" that analyzes inconsistencies among multiple generations rather than simply aggregating them. This approach shows improvements in both reasoning accuracy and confidence calibration by enabling models to learn from discrepancies in their reasoning pathways. Recent advances include Speculative Ensemble methods that accelerate LLM by generating and verifying tokens in parallel without sacrificing performance (Fu et al., 2025), and diversity-optimized ensemble techniques that use focal diversity metrics to select optimal sub-ensembles from larger model pools (Tekin et al., 2024).

Despite these advances, ensemble methods remain fundamentally constrained by the systematic biases of their constituent validators. Standard majority voting is insufficient for addressing the low True Negative Rates that characterize LLM judges, as demonstrated by us in §4. While minority-veto strategies can improve performance, they still cannot overcome the inherent limitations of LLM judges.

**Calibration and Bias Correction**. Calibration techniques aim to ensure model confidence scores accurately reflect correctness probability. Traditional approaches like temperature scaling have been extended through adaptive means, but these primarily address general overconfidence rather than systematic evaluation biases (Guo et al., 2017). In the LLM domain, selective generation based calibration methods leverage consistency across multiple generations to identify unreliable results (Lin et al., 2023).

However, existing calibration methods focus on confidence scaling rather than addressing the fundamental issue we identify: systematic tendencies of LLM judges to agree with invalid outputs. Recent work on post-hoc reward calibration attempts to estimate bias terms using local reward averaging approaches (Huang et al., 2024b), but these methods do not specifically target the low TNR problem that compromises evaluation reliability. Our regression-based approach differs by explicitly modeling generator-validator interactions and using ground-truth data to calibrate for specific biases, representing a shift from general confidence calibration to targeted bias correction for evaluation systems.

The growing recognition of these limitations has led to increased interest in developing evaluation methods that combine automated scalability with human judgment reliability. Our work addresses these limitations through explicit bias modeling and correction.

