# OpenReview forum: "Beyond Consensus: Mitigating the Agreeableness Bias in LLM Judge Evaluations"
_ICLR.cc/2026/Conference — Submitted to ICLR 2026_

### Official Review · Reviewer_cH21 · 2025-10-28

**Soundness:** 2
**Presentation:** 3
**Contribution:** 2
**Rating:** 4
**Confidence:** 4

**Summary:**

The paper identifies the positive bias problem in LLM-as-a-judge, i.e., LLM verifiers have high TPR while low TNR. The paper first proposes minority-veto strategy, an improvement upon the majority-voting method that marks the response as invalid when the number of individual verifiers that output "invalid" reaches a certain threshold. This threshold strikes a balance between TPR and TNR. In addition, minority-veto strategy is robust to missing data, a problem for majority-voting. Then the paper proposes the regression-base approach, another bias correction strategy that assumes access to a small proportion of human annotated data. Based on the assumptions that verifiers' TPR/TNR are independent of the generator, this strategy learns the verifiers' TPR and TNR with the accessible human data, which can be used to predict the precision of a new generator. Finally, the paper creates a new dataset to test the two approaches.

**Strengths:**

The paper formalizes the positive bias problem as the gap between TPR and TNR, and provides empirical evidence for it. The paper provides two novel solutions, one as a fix for the popular majority-voting strategy, the other based on regression.

**Weaknesses:**

1. Weakness in quality: All experiments in the paper are conducted on a self-created dataset, without any experiment results on public datasets. In addition, the dataset contains only 366 problems, so the evaluation result suffers from high variance.
2. Weakness in clarity:
    - The term "precision" in the paper is confusing, especially when appearing together with TPR and TNR. In line 142, "precision" is defined as "the proportion of valid feedback among all successfully generated items". This violates the [usual definition](https://developers.google.com/machine-learning/crash-course/classification/accuracy-precision-recall) of "precision" (TP/(TP+FP)). In fact, "precision" should be renamed as "accuracy" according to common conventions.
    - The regression-based approach is based on the strong assumption that the TPR and TNR of the verifiers does not change with generators. In practice, this assumption does not hold, as Eq. (2) and (3) cannot be minimized simultaneously.
3. Weakness in significance: The two strategies proposed in the paper are completely independent, and they cannot be combined together. Therefore, the paper can be viewed as the combination of two separate works.
4. Weakness in novelty: The minority-veto approach is essentially choosing the decision threshold in the classical precision-recall trade-off , which is widely discussed in ML research.

**Questions:**

1. How do the proposed methods perform in publicly accessible datasets?
2. As a sanity check, how do the methods perform on tasks with verifiable answers? Benchmarks including GSM8K, MATH-500, etc. These benchmarks add reliability to the proposed methods, as the answers are objective and the benchmarks have more data.

---

> ### Author Response · Authors · 2025-11-21
>
> We thank the reviewer for their positive feedback on our formalization of the positive bias problem and novelty of the proposed solutions. Here are some clarifications on the weaknesses highlighted
>
> ### 1. On Dataset and Generalizability:
>
> We wish to clarify that our dataset is not created from scratch but was built upon an existing public benchmark by Sahai et al., 2023. A significant effort (over 200 person hours) went into extending it for 5 additional generators and will be released for further research.
>
> Regarding benchmarks like GSM8K or MATH-500, we respectfully argue that these address a fundamentally different problem class where, as pointed out by reviewer, verifiable answers exist (via string/number matching), rendering LLM-as-a-judge unnecessary. As stated in our introduction, our work instead focuses on open-ended problems where correctness is subjective and not verifiable by a simple oracle. Creating ground-truth human annotated data for such tasks is costly, which explains the scarcity of such public datasets. Our deep dive into this complex domain allowed us to uncover and quantify the "agreeableness bias" (low TNR) often overlooked in prior work.
>
> Regarding dataset size: while it contains 366 programs, each incorrect program elicits multiple feedback items, resulting in thousands of data points per generator (e.g., 1,230 items for Gemini 1.5 Pro, 1,046 for Deepseek 2.5; see Table 1). This scale is comparable to benchmarks like MATH-500.
>
>
> ### 2.1  On the Definition of "Precision":
>
> Our use of "precision" for the generator (g_i) is indeed consistent with the standard definition. In our problem setting, the generator's task is to identify issues in buggy code and provide feedback.
>
> That is:
> - True Positive (TP) is generated feedback deemed valid by human experts
> - False Positive (FP) is an invalid feedback item
> - Positive prediction (TP+FP): is any feedback item produced by a generator
>
> Thus, our definition "the proportion of valid feedback among all successfully generated items," is exactly TP / (TP + FP). Please refer to Table 1 for concrete examples. We use TPR and TNR exclusively to evaluate the *validators* (LLM-as-a-judge), not the generators, and will revise text to make the distinction explicit.
>
> ### 2.2 On the Regression Model's Assumption:
>
> We acknowledge that assuming static TPR/TNR for validators is a simplification (as noted in Limitations, Sec. 7). However, our empirical results (Figure 6) demonstrate that this model is highly effective in practice, significantly outperforming other methods by calibrating for systematic bias, even if it doesn’t capture every generator-validator nuance.
>
> Regarding loss minimization, we clarify that we do not minimize the prediction loss (L _pred) and calibration loss (L_cal) independently. The total loss is their weighted sum: L = L_pred + L_cal. This joint minimization allows the calibration loss to act as a regularizer, anchoring the optimization with limited ground-truth data. This guides the model away from solutions reflecting validator bias and towards solutions aligning with human judgment.
>
> ### 3. On the Weakness in Significance:
>
> The reviewer notes that our two proposed strategies are independent. We view this as a strength. We offer two different solutions for the same problem for practitioners:
> - If human annotated ground truth is available: Regression is the optimal choice. It leverages this small (s=1 or 2 generator data) one-time investment to explicitly calibrate for validator bias, achieving a 2x reduction in maximum absolute error.
> - If no ground truth is available: The Minority Veto offers a robust, computationally efficient simple heuristic that significantly outperforms standard majority voting and individual judges.
>
> ### 4. On the Novelty of Our Contributions:
>
> Regarding the novelty of Minority Veto, while thresholding is a common ML technique, our contribution lies in its application to address the specific "agreeableness bias" (high TPR, low TNR) of LLM judges. To the best of our knowledge, we are the first to show that empowering a dissenting minority of LLM judges is significantly more effective and robust to data quality issues than standard majority consensus and state-of-the-art individual LLM judges. We are unaware of prior work applying this strategy to LLM evaluations and welcome references if we have overlooked any.
>
> ### Regarding the questions:
> As detailed in point 1, our work extends an existing public benchmark to a scale comparable to standard datasets. Alternative benchmarks (e.g., GSM8K, MATH-500) do not require LLM-as-a-judge validators, as the correctness of the LLM generated output can be determined via simple exact-match oracles. Our method is specifically designed for open-ended, subjective tasks where such oracles do not exist due to the presence of potentially infinitely many valid answers. Hence, LLM judges and their inherent biases are one of the only scalable alternatives to human annotation.

---

> > ### Comment · Reviewer_cH21 · 2025-11-25
> >
> > Thanks for the authors' response. It resolves most of my concerns. However, the narrow domain of the experiment is still an issue. Take the regression method as an example. Though it is proposed to solve open-ended tasks, the approach applies directly to tasks with verifiable answers. Therefore, it would be helpful to test the regression method on one or two math/logical tasks to show the robustness approach. As a result, I choose to keep the score for now.

---

> ### Author Response · Authors · 2025-11-28
>
> Thank you for the positive response and we are glad our previous reply addressed most of the  concerns. We believe there may be a misunderstanding regarding the scope of our work, and hope the clarification below fully resolves the remaining concern regarding the generalizability of our approach.
>
> ### Our problem class: open-ended tasks with verifiable answers
>
> Our work specifically targets tasks where (1) multiple distinct outputs can be correct, yet (2) experts can objectively label correctness with high inter-annotator agreement. This is precisely where LLM-as-a-judge is most needed, as simple string matching or automated oracles are insufficient. For general open-ended tasks without verifiable answers (e.g., creative writing), accuracy itself is ill-defined, and such tasks fall outside our scope.
>
>
> ### Why math/logic tasks are not representative
>
> Math and logic tasks, while important, typically have a unique correct answer (ground truth) that can be automatically verified (e.g., symbolic solvers, unit tests). Because correctness is deterministically computable, such tasks do not require LLM judges and are therefore not representative of the problem class we address. Such datasets would not meaningfully stress-test our approach.
>
> In contrast, programming feedback (and more broadly code explanation) exemplifies the exact setting of our approach. Feedback can be expressed in infinitely many valid ways, yet correctness is still objectively verifiable by experts. This makes it both challenging and highly apt domain for LLM-as-a-judge research.
>
>
> ### On dataset availability
>
> To the best of our knowledge, no other existing benchmark provides expert-validated absolute correctness labels for open-ended but verifiable tasks. Extending the public dataset from Sahai et al. (2023) required 200+ person-hours of expert annotation (about 4 weeks). Repeating this for another domain during rebuttal is infeasible. Moreover, as acknowledged in our paper, proving generalizability across *all* domains is beyond our current scope. However, if the reviewer can point us to any publicly available dataset meeting our criteria, we would gladly evaluate our methodology on it.
>
> ### The significance of our domain
>
> We respectfully emphasize that code explanation and feedback is not a narrow domain. it is one of the most active application areas for LLMs today. Students rely on LLM-generated explanations to learn, universities and MOOCs are integrating AI tutors into curricula, and industry platforms depend on automated feedback for developer training. A rigorous dataset enabling absolute evaluation in this domain fills a significant gap.
>
> For practitioners working in a similar domain for which our technique is shown to work (code feedback/explanation), our work offers substantial improvements over existing llm-as-a-judge evaluation methods.
>
>
> ### Summary of contributions
>
> Our paper offers: (1) identifying an underexplored judge bias and empirically presenting it (low TNR), (2) two robust mitigation techniques, one requiring no ground truth (minority veto), another achieving 2x improvement with minimal annotation (regression), and (3) the first high-quality dataset, to the best of our knowledge, for absolute precision evaluation on open-ended code feedback.
>
> We would also like to refer to our latest comment to [reviewer Ri2p](https://openreview.net/forum?id=3IsAoRHi9n&noteId=542OZ3Yb7d) for a detailed discussion on dataset & generalisability along with examples.
> >  If similar public human-annotated datasets existed for our problem setting, we would gladly run additional experiments (to demonstrate robustness & generalisabiilty).
>
> We hope this clarifies the remaining concern and would like to respectfully request the reviewer to reconsider their score in light of these.

---

### Official Review · Reviewer_Ri2p · 2025-10-29

**Soundness:** 1
**Presentation:** 2
**Contribution:** 1
**Rating:** 2
**Confidence:** 4

**Summary:**

This paper studies a practical and timely problem: the reliability of LLMs used as validators (LLM-as-a-judge) for open-ended tasks where correctness is non-binary (here: automated feedback for incorrect high-school Python programs). The authors introduce and analyse the “agreeableness bias”, validators show very high True Positive Rate (TPR ≳ 96%) but very low True Negative Rate (TNR ≲ 25%), causing systematic over-estimation of a generator’s precision. Using a dataset of 366 buggy programs, feedback generated by 14 LLMs, and human annotations for 6 generators, the paper (i) quantifies the bias (Figures 1, 3–4, Table 1), (ii) shows limits of simple majority voting (sensitivity to missing outputs and a floor imposed by validator TNR), (iii) proposes a Minority-Veto ensemble that improves robustness to missing values, and (iv) presents a regression-based calibration that models each validator’s TPR/TNR and each generator’s precision, using a small calibration set of human-annotated generators. The regression approach, when calibrated with up to 5 annotated generators, reduces maximum absolute error to ~1.4% (Figure 6), outperforming ensembles.

**Strengths:**

* Simple, effective ensemble method. The Minority-Veto rule is conceptually simple, robust to missing values, and empirically outperforms plain majority voting after minimal calibration.

* Reproducibility intent & dataset release. The authors commit to releasing code and the dataset (14 LLM outputs × 366 tasks), which will be useful for follow-up work.

**Weaknesses:**

* Single domain study; limited evidence of generality. All experiments are on code-feedback generation (366 programs). The authors acknowledge this; however, it remains unclear how agreeableness and the regression correction behave on other open-ended domains (summaries, creative writing, dialogue, medical advice) where human agreement patterns differ.

*The paper compares the proposed methods mainly against simple majority voting and a random baseline, but omits other relevant approaches from related domains such as advanced aggregation schemes (e.g., weighted or probabilistic voting), existing calibration or de-biasing techniques used in LLM judgment tasks, or classical ensemble reliability models. Without broader comparisons, it is difficult to gauge the relative contribution and novelty of the proposed ensemble and regression methods beyond these straightforward baselines.

**Questions:**

N/A

---

> ### Author Response · Authors · 2025-11-21
>
> We thank the reviewer for their summarization of our paper and acknowledge their comment on “single domain study”, which we have highlighted in our limitations section as well.
>
> However, we are confused by the low scores for “soundness” and “contribution”, which seem inconsistent with their summary and identified weakness. This is perhaps due to a misunderstanding of our paper's scope and a misreading of our contributions, which we would like to re-emphasize below.
>
> ### 1. On the Single Domain Study:
>
> We respectfully disagree that our focus on a single domain is a critical weakness. As stated in our paper (Section 2), this choice was deliberate and necessary.
> - Scarcity of Data: High-quality, human-annotated datasets for open-ended, subjective tasks with multiple correct answers are very rare due to the immense effort required to create and annotate them. Our work builds on one of the few such public datasets we are aware of and extends it significantly by investing over 200 person-hours of additional effort.
> - Depth over Breadth: This deep dive into a single, complex domain is a core strength of our paper, as it allowed us to move beyond vague notions of "positive bias" and be the first to empirically quantify it as a specific, critical imbalance between high TPR and low TNR. This is a critical insight, which forms the foundation of our entire work, and would likely be obscured in a broader, less-controlled study across multiple domains with varying nuances.
> - Generalizable Methodology: Our primary contribution is methodology for mitigating this bias, which the reviewer has acknowledged as effective. The minority veto ensemble and regression framework is, by design, general purpose that researchers in other domains, whether *summarization or creative writing or medical advice*, can now adopt and benchmark LLMs with, provided they are willing to make a similar one-time investment in calibrating their own domain-specific evaluation pipeline. Once this small calibration set is ready, no additional human effort is required to benchmark new un-annotated generators.
>
> ### 2. On the Comparison to Baselines:
>
> We would like to reject assertions that we omitted relevant comparisons. The reviewer's claim that we only compare against "simple majority voting and a random baseline" suggests a misreading of our paper. We respectfully request the reviewer to re-examine our Related Work (Section 6) and our methodology.
>
> - Advanced Aggregation Schemes: We explicitly discuss and differentiate our work from classical latent trait models. In Section 6, we state: *"Our regression framework is conceptually related to classic models for aggregating noisy labels, such as the EM Algorithm (Dawid & Skene, 1979) and Item Response Theory (IRT) (Lord, 2012)."* We argue why our approach is more suitable for estimating absolute precision, whereas IRT is designed for relative ranking.
> - Calibration and De-biasing Techniques: Our entire regression framework is in fact a novel de-biasing and calibration technique. We dedicate a full section to this method (Section 5), which directly models and corrects for the specific TPR/TNR imbalance we identified. In Section 6, we compare this to existing calibration literature, and note that *"existing methods typically address general overconfidence rather than the specific, systematic low-TNR problem that we identify. Our regression-based approach differs by using a ground-truth calibration set to explicitly model and correct for this bias..."*
>
> We appreciate the reviewer indicating the vast literature on aggregation and calibration. We believe our work is positioned within the context of these advanced methods and our paper already contains the necessary comparisons, to the best of our knowledge.
>
> Having said that, we are open to strengthening our experimental results. If the reviewer could kindly suggest specific algorithms that we may have overlooked, we would be happy to compare against them and update our draft.

---

> > ### Comment · Reviewer_Ri2p · 2025-11-24
> >
> > Thanks for the author's response. It addressed some of my concerns about baselines used in this work. However, I still believe using more datasets in different domains is a critical point in verifying the generalization of the proposed methods. Below, I list several potential judgment datasets, and if the authors could conduct additional experiments on some of them, or clarify why they (and other reward-bench) are not applicable in the scenario this paper focused on, I am open to increasing my score.
> >
> > https://huggingface.co/datasets/allenai/reward-bench
> >
> > https://huggingface.co/datasets/lmarena-ai/PPE-Human-Preference-V1
> >
> > https://huggingface.co/datasets/lmsys/mt_bench_human_judgments
> >
> > https://huggingface.co/datasets/prometheus-eval/Feedback-Bench
> >
> > https://huggingface.co/datasets/prometheus-eval/Preference-Bench

---

> > > ### Author Response · Authors · 2025-11-28
> > >
> > > We appreciate the reviewer's continued engagement and are grateful that our earlier response addressed many of their concerns. The follow-up raises an important question about cross-domain generalization. We see this as an opportunity to further clarify the scope and contributions of our paper.
> > >
> > > After carefully reviewing each suggested dataset, we found that, unfortunately, none of them fall within our problem definition. As stated in our introduction, our paper focuses on **absolute benchmarking**, where our goal is to determine the actual accuracy of a LLM, rather than relative benchmarking, where existing works try to determine which of 2 LLMs is better.
> > >
> > > > "benchmarking is significantly more challenging for a new class of open-ended problems where multiple distinct outputs can be correct"
> > >
> > > ---
> > >
> > > ## Our dataset
> > >
> > > Our dataset (to be released publicly) is designed to meet two requirements essential for absolute precision benchmarking:
> > >
> > > 1. **Open-ended solution space:** For a given error in the code, infinitely many valid feedback responses may exist. However, it is possible to determine unambiguously whether a given feedback is valid
> > > 2. **Binary ground truth with high inter-annotator agreement:** Each feedback item is labeled valid/invalid by human experts, enabling measurement of precision, TPR, and TNR.
> > >
> > > The structure of our dataset is:
> > >
> > > | Generator Output (Model A) | | Ground Truth | Validator Output | |
> > > |---|---|---|---|---|
> > > | **Line** | **Feedback** | **Human** | **Model B** | **Model C** |
> > > | 5 | Instead of printing the result, your function should return it instead | Valid | Valid | Invalid |
> > >
> > > ---
> > >
> > > ## Why the suggested datasets are incompatible
> > >
> > > **Four of five datasets are designed for relative preference evaluation:**
> > >
> > > - [reward-bench](https://huggingface.co/datasets/allenai/reward-bench)
> > > - [PPE-Human-Preference-V1](https://huggingface.co/datasets/lmarena-ai/PPE-Human-Preference-V1)
> > > - [mt_bench_human_judgments](https://huggingface.co/datasets/lmsys/mt_bench_human_judgments)
> > > - [Preference-Bench](https://huggingface.co/datasets/prometheus-eval/Preference-Bench)
> > >
> > > Their structure looks like:
> > >
> > > | Generator Output (Model A) | Generator Output (Model B) | Human Preference |
> > > |---|---|---|
> > > | From your description, it seems like you are trying to listen for changes in the `chrome.storage`… | The issue seems to be related to how the `chrome.storage.onChanged` event is being handled… | model_a |
> > >
> > > Here, human annotations reflect which output is preferred, not whether either output is objectively valid. Ranking Model A > Model B does not imply Model A's output is valid and Model B's is invalid. Both A and B could be valid or invalid. Hence, these datasets cannot support our problem setting.
> > >
> > > **The remaining dataset uses subjective scoring:**
> > >
> > > - [Feedback-Bench](https://huggingface.co/datasets/prometheus-eval/Feedback-Bench)
> > >
> > > The structure looks like:
> > >
> > > | Subjective Task | Generator Output (Model A) | Score (1-5) |
> > > |---|---|---|
> > > | Could you help me rewrite the following paragraph in a more engaging, humorous way?… | Meet Sam, our furry hero… | 1/5 |
> > >
> > > This dataset uses subjective 1-5 Likert scoring for stylistic or creativity-based tasks where inter-annotator agreement is inherently low. Additionally, annotations exist for only a single generator, making it impossible to compute TPR/TNR for validators.
> > >
> > > While we genuinely appreciate these suggestions, these are well-constructed datasets for their specific problem setting. If public human-annotated datasets were available for our problem setting, we would gladly run additional experiments.
> > >
> > > ---
> > >
> > > ## The significance of our domain
> > >
> > > We emphasize that programming feedback or code explanation is not a narrow domain. It is central to how LLMs are deployed in education, training, and software engineering. Students rely on LLM-generated explanations to learn, universities and MOOCs are integrating these tools into teaching pipelines, and industry platforms are investing heavily in AI-driven code tutoring and review. Evaluating LLMs' ability to generate correct, and actionable programming feedback is a foundational use case.
> > >
> > > Yet, to the best of our knowledge, **no existing benchmark provides high-agreement, expert-validated absolute correctness labels for this domain**. To solve this, we spent 200+ person-hours (~4 weeks) to annotate 5000+ feedback items generated. Investing this effort again to annotate a new dataset to prove cross-domain generalizability is infeasible during the rebuttal period and beyond the scope of our work.
> > >
> > > Our methodology and dataset provide the community with the first rigorous tool to evaluate LLM judges on open-ended code feedback. We believe this makes a meaningful and timely contribution, and we leave applicability to other domains as an important future extension of our work.

---

### Official Review · Reviewer_u2Jz · 2025-10-30

**Soundness:** 3
**Presentation:** 2
**Contribution:** 2
**Rating:** 4
**Confidence:** 4

**Summary:**

The paper presents the "agreeableness bias” of LLM-as-a-judge on the task of programming feedback, described as having high TPR and low TNR. The authors show that strong models also exhibit this bias. But an ensemble method like majority vote and a proposed variant of it can mitigate it. Moreover, the paper presents a regression-based method that uses the full judgment data and human-annotated calibration data to further reduce this bias.

**Strengths:**

1. Good motivation, utilizing previous work findings on positive bias, and the need for improved open-ended task evaluation.
2. A good number of models in the experiments.
3.

**Weaknesses:**

1. The motivation and experimental setup are not aligned: a. The task of "programming feedback" is different then LLMaaJ. b. The experiment data is very narrow and not general enough to frame the paper as a general LLMaaJ results. The authors recognize this in a few places, but their response is not sufficient. This is a core problem of the paper.
2. Some parts are not clear enough. The predicted and estimated precision, and if it's just the LLM precision, why and how the close agreement between Validator Mean and ground truth aligns with the positive bias.
3.  “invalid” responses can exist, but using established methods and best practices like structured output, etc., can reduce them. Moreover, it can stem from choosing relatively old models.
4. Minority Veto strategy seems very specific to your setup, and ignores the quality of each judge. It can just be the MV of the top-4.
5. The regression-based method is not feasible, requires many judges, and human-annotated data.
6. Going back to the motivation, the paper doesn't address the motivation it raises or provide a solution that is better or more accurate than the current method.

**Questions:**

The paper needs to be framed in the context of the target domain or adapted to general data suited for the assessment of LLMaaJ. The writing needs to be improved and clearer to make the paper's point about "agreeableness bias” and its mitigation really come across.

---

> ### Author Response · Authors · 2025-11-21
>
> We thank the reviewer for their insightful feedback and offer clarifications below
>
> ### 1. On the Scope of the Experimental Setup:
> Our core contribution is a novel domain agnostic methodology to determine and mitigate the LLM judge bias. We demonstrate that "agreeableness bias” exists in the critical domain of “program feedback” or “code explanation”, and that our methodology can help address it.
>
> While we do not claim universal generalization, there is no guarantee that applying it to another subjective task implies it would work on all other domains. Exploring this extension is left as future work as datasets for open-ended problems with multiple valid solutions are rare, and it is simply too expensive to manually annotate the generator outputs by experts to measure the absolute error rate against ground truth.
>
> ### 2. On Clarity (Predicted vs. Estimated Precision and Positive Bias):
>
> We apologize for the confusion and will revise the text. Here, "predicted" refers to precision obtained from individual LLM judges, while "estimated" refers to the final computation by our ensemble/regression methods.
>
> Regarding the “validator mean”, as seen in Figure 2, while the mean aligns with ground truth, the variance is unacceptably high. The mean is a misleading aggregate of opposing biases: most validators overestimate precision (due to low TNR), while a few (e.g., Gemini) are strict and underestimate it. Our work corrects these underlying biases by explicitly modeling each validator's TPR and TNR..
>
> ### 3. On Invalid Validator Responses:
>
>
> While structured outputs can reduce invalid responses to some extent, such as malformed JSONs, it doesn’t resolve the full spectrum of missing value cases. As detailed in Appendix B.2, validators may hallucinate generator’s response or refer to non-existent line numbers, leading to labels that cannot be matched against generator's feedback. Issues like these cannot be addressed through structured outputs.
>
> Notably, Minority Veto (MV) ensemble and regression methodology are largely robust to such invalid responses and missing values.
>
> ### 4. Minority Veto (MV) generalizability:
> We respectfully disagree that a "top-4" judge ensemble is feasible. Figure 4 shows there is no stable "top-k" set of judges with a consistent TPR/TNR trade-off. Moreover, a fixed subset would be brittle and generalize poorly. In contrast, our validator-agnostic MV strategy leverages the "wisdom of the crowd", empowering a dissenting minority to flag errors which majority misses. Figure 6 confirms this strategy is significantly more robust & generalizable, and resilient to missing data, compared to standard voting.
>
> ### 5.On the Feasibility of the Regression Method:
> We strongly argue that our method is highly practical and directly addresses the scalability bottleneck of LLM benchmarking. The dominant cost here is human annotation, which we reduce to a **one-time** effort. Our results show that annotating just a single generator (s=1) yields significant gains, and evaluating any number of new models requires **zero** additional human effort.
>
> Running multiple LLM validators is a trivial, parallelizable computational cost compared to the unscalable alternative of manual expert review.
>
> ### 6. On Addressing the Stated Motivation:
>
> We are confident that our work directly addresses unreliability of individual LLM judges caused by "agreeableness bias". We provide two solutions significantly more accurate and reliable than the state-of-the-art (single LLM judge) on programming feedback:
> - Minority Veto ensemble reduces the max estimation error from 17.5% (worst single judge) to 2.8%.
> - Regression further reduces this error to just 1.4%, a 2x improvement over the best ensemble.
>
> ## Regarding the questions:
>
> We note the comment on the writing issue and will strive to improve the clarity of our paper.
>
> However, we would respectfully disagree that our domain is not representative. Existing "general" benchmarks (e.g., MMLU, HumanEval) rely on trivial string-matching or straight-forward evaluation oracles. These tasks do not require complex reasoning to evaluate. Our programming feedback task in fact represents the actual frontier of LLM benchmarking, containing open-ended problems with multiple valid solutions. This is exactly where LLM judges are most required, and where they fail spectacularly due to their "agreeableness bias".
>
> Regarding adopting to general data: High-quality, human-annotated ground truth for open-ended tasks are rare. We invested over 200 person-hours to expand on an existing public benchmark specifically because no "general" alternatives exists, to the best of our knowledge. However, we acknowledge that we may have missed relevant resources. We would be grateful for specific dataset suggestions with similar characteristics (open-ended, multiple valid answers, human-annotated), and we would be happy to incorporate them into our evaluation.

---

### Meta-Review · Area_Chair_A6ck · 2026-01-05

**Summary:**

This paper sets out to establish that LLMs have an "agreeablness" bias whereby they identify too many responses as of good quality. The paper presents a method for improving the detection rates of invalid examples. The paper evaluates in the context of code generation tasks, where there are many solutions but they can be objectively graded.

Strengths:

- Points out an important flaw in LLM-as-a-judge

- Simple "minority veto" method for mitigating that flaw

Weaknesses:

The main weakness is the limited domain nature.  Although the paper claims to focus on open-ended problems, I think the current study is too limited to make the claims it makes.  LLM-as-a-judge is used in practice for things like writing quality assessment, rubric-based grading of deep research responses, safety, etc.; I don't think it's clear that the claims from this domain generalize to those. I recognize this is very challenging to do research on, but I do think the limitation makes the claims of this work not hold.

The authors make two claims with regard to this, which I will push back on.

> Existing "general" benchmarks (e.g., MMLU, HumanEval) rely on trivial string-matching or straight-forward evaluation oracles. These tasks do not require complex reasoning to evaluate. Our programming feedback task in fact represents the actual frontier of LLM benchmarking, containing open-ended problems with multiple valid solutions.

I agree that many papers have studied these domains and framed them as general.  Evaluating a pre-trained LLM on MMLU or HumanEval is a way to benchmark its general capabilities, and these scores often correlate with scores on other domains.  That is why these problems can be treated as general: because they have been established in the community to have this flavor (whether that process is rigorous enough or not is a separate question).

Unfortunately, none of this means that the programming feedback task is the frontier of LLM benchmarking. There's a false dichotomy here that ignores other existing work.

> High-quality, human-annotated datasets for open-ended, subjective tasks with multiple correct answers are very rare

I would encourage the authors to look at one of the following tasks/datasets, all of which feature human-annotated data for open-ended, subjective tasks:

- SummEval ( https://arxiv.org/pdf/2007.12626 )

- Long-form QA ( https://arxiv.org/pdf/2305.18201 )

- DOLOMITES ( https://arxiv.org/pdf/2405.05938 ) see Figure 5

- MT-Bench ( https://github.com/lm-sys/FastChat/tree/main/fastchat/llm_judge#agreement-computation )

Doubtless there are others; these are just what I can list off the top of my head.

**Reviewer Concerns:**

The reviewers had several concerns:

1. Limited domain nature (all 3 reviewers): not resolved, see my comment above
2. Lack of clarity, definition of precision: resolved, I don't see this as a major issue
3. Omitting comparison to relevant baselines: the authors push back on this, I think the issue is partially resolved. The method itself is quite simple, which is nice,
and I'm not sure that many complex methods need to be compared to here.
4. Significance/novelty: again, the method is simple, and that's okay. So I think this is resolved.

**Reviewer Scores:**

I believe the response partially addresses the issues, but does not address a fundamental weakness of the work as described above.

---

### Decision · Program_Chairs · 2026-01-26

Reject